# Whole brain and deep gray matter structure segmentation: Quantitative comparison between MPRAGE and MP2RAGE sequences

Amgad Droby[1,2,3]*, Avner Thaler[1,2,3], Nir Giladi[1,2,3], R. Matthew Hutchison[4], Anat Mirelman[1,2,3], Dafna Ben Bashat[2,3,5], Moran Artzi[2,3,5]

**1** Laboratory for Early Markers of Neurodegeneration (LEMON), Neurological Institute, Tel Aviv Sourasky Medical Center, Tel Aviv, Israel, **2** Sackler School of Medicine, Tel Aviv University, Tel Aviv, Israel, **3** Sagol Brain Institute, Tel Aviv Sourasky Medical Center, Tel Aviv, Israel, **4** Biogen Inc., Cambridge, MA, United States of America, **5** Sagol School of Neuroscience, Tel Aviv University, Tel Aviv, Israel

* amgadd@tlvmc.gov.il

**Data Availability Statement:** De-identified data set supporting this work are available from the upon institutional review board (IRB) at Tel Aviv

## Abstract

### Objective

T1-weighted MRI images are commonly used for volumetric assessment of brain structures. Magnetization prepared 2 rapid gradient echo (MP2RAGE) sequence offers superior gray (GM) and white matter (WM) contrast. This study aimed to quantitatively assess the agreement of whole brain tissue and deep GM (DGM) volumes obtained from MP2RAGE compared to the widely used MP-RAGE sequence.

### Methods

Twenty-nine healthy participants were included in this study. All subjects underwent a 3T MRI scan acquiring high-resolution 3D MP-RAGE and MP2RAGE images. Twelve participants were re-scanned after one year. The whole brain, as well as DGM segmentation, was performed using CAT12, volBrain, and FSL-FAST automatic segmentation tools based on the acquired images. Finally, contrast-to-noise ratio between WM and GM ($CNR_{WG}$), the agreement between the obtained tissue volumes, as well as scan-rescan variability of both sequences were explored.

### Results

Significantly higher $CNR_{WG}$ was detected in MP2RAGE vs. MP-RAGE (Mean ± SD = 0.97 ± 0.04 *vs.* 0.8 ± 0.1 respectively; *p*<0.0001). Significantly higher total brain GM, and lower cerebrospinal fluid volumes were obtained from MP2RAGE vs. MP-RAGE based on all segmentation methods (*p*<0.05 in all cases). Whole-brain voxel-wise comparisons revealed higher GM tissue probability in the thalamus, putamen, caudate, lingual gyrus, and precentral gyrus based on MP2RAGE compared with MP-RAGE. Moreover, significantly higher WM probability was observed in the cerebellum, corpus callosum, and frontal-and-temporal regions in MP2RAGE vs. MP-RAGE. Finally, MP2RAGE showed a higher mean percentage of change in total brain GM compared to MP-RAGE. On the other hand, MP-RAGE

Sourasky Medical Center (Tel: +972-3-6974003; FAX: +972-3-6973974; Email: anatmi@tlvmc.gov.il) upon reasonable request by researchers who meet the criteria for access to confidential data.

**Funding:** This study was funded by Biogen Inc., who also provided support for this study in the form of salary for RMH. The specific roles of these authors are articulated in the 'author contributions' section. The funders had no role in study design, data collection and analysis, decision to publish, or preparation of the manuscript.

**Competing interests:** The authors have read the journal's policy and the authors of this manuscript have the following competing interests: RMH is a paid employee of Biogen Inc. and owns stock in the company. NG serves as a member of the Editorial Board for the Journal of Parkinson's Disease, as a consultant to Sionara, Accelmed, Teva, NeuroDerm, Intec Pharma, Pharma2B, Denali, and Abbvie, receives royalties from Lysosomal Therapeutics (LTI), payment for lectures at Teva, UCB, Abbvie, Sanofi- Genzyme, Bial, and Movement Disorder Society, received research support from the Michael J Fox Foundation, the National Parkinson Foundation, the European Union 7th Framework Program and the Israel Science Foundation as well as from Teva NNE program, Biogen, LTI, and Pfizer. There are no patents, products in development or marketing products to declare. This does not alter our adherence to PLOS ONE policies on sharing data and materials. AD, AT, AM, DBB, and MA declare no competing of interests pertaining to this work.

demonstrated a higher overtime percentage of change in WM and DGM volumes compared to MP2RAGE.

## Conclusions

Due to its higher CNR, MP2RAGE resulted in reproducible brain tissue segmentation, and thus is a recommended method for volumetric imaging biomarkers for the monitoring of neurological diseases.

## Introduction

Magnetic resonance imaging (MRI) is a commonly used method for investigating the neural hallmarks and the monitoring of many neurologic and neuropsychiatric diseases [1]. Assessing the whole brains' grey matter (GM) and white matter (GM) tissue, as well as deep grey matter (DGM) structures, can be of high importance for the diagnosis and therapy selection in many neurological and neuropsychiatric conditions [2]. This can be achieved mainly by segmenting the brain structures based on T1-weighted, T2-weighted, proton density (PD), magnetic transfer (MT), or multi-contrasts [3,4].

Magnetization Prepared Rapid Gradient Echo (MP-RAGE) [5] is the predominantly used sequence for obtaining 3D T1-weighted MR images of the human brain [3]. MP-RAGE consists of a non-selective (180°) inversion pulse, which inverts the net magnetization (M), allowing M to regrow via T1 relaxation mechanisms over the inversion time interval (TI), during which, the signal is acquired using a spoiled gradient echo (GRE) (Turbo-FLASH) with a low flip angle.

At high (1–3 Tesla) and ultra-high ($\geq$7 Tesla) magnetic fields, the increased in-homogeneities of transmitting and receiving magnetic fields (B1) create intensity variations in the acquired images leading to reduced contrast, affecting segmentation results and accuracy of tissue -classifications [3]. This may even be more pronounced in DGM which have shorter T1 values compared to higher cortical regions, leading to decreased tissue contrast between GM and neighboring white matter (WM) [6,7]. Such classification in-accuracies could result in poorer sample size calculations and decreased power in clinical trials.

MP2RAGE is becoming readily available across several MR vendors. The sequence is a variation of the MP-RAGE, which utilizes two Turbo-FLASH GRE readouts acquiring 2 volumes after each inversion pulse. The first TI (TI1 $\approx$ 700ms) produces a T1-weighted image with the gray matter nulled at the center of $k$-space, and the second TI (TI2 $\approx$ 2500ms), combined with small FAs ($\alpha$ = 4–5°), and long TR (~5000ms), produces a second image with spin-density-weighted contrast. By combining both images from the first and second readouts, $T2^*$ and B1 inhomogeneity effects are largely canceled out resulting in a T1-weighted unified (UNI) image with superior GM-to-WM contrast compared to MP-RAGE, allowing a reproducible quantification of DGM structures volumes across subjects [8,9].

Several studies applying MP2RAGE have reported enhanced contrast-to-noise ratio (CNR), as well as a good inter-and-intrasubject agreement between the acquired T1-maps [3,10], yielding larger measured volumes for several DGM structures, as well as more sensitive disease burden markers in neurological conditions, were obtained when assessed based on MP2RAGE images compared to MP-RAGE [9,11]. However, the quantification of tissue volumes does not rely on the sequence acquisition parameters alone. Factors related to the applied algorithm used for segmentation also contribute to the reliability of the obtained volumetric values

[12,13]. Therefore, in this study, we aimed to systematically examine the patterns of discrepancies and quantitatively assess the agreement between tissue classification obtained by different widely available, automatic segmentation tools relying on different algorithms that are commonly used for the assessment of whole-brain tissue, as well as DGM volumes based on MP2RAGE compared to MP-RAGE images.

## Materials and methods

### Study participants

This study was part of the ongoing BEAT-PD study taking place since 2017 at the Tel Aviv Sourasky Medical Center (TASMC). Data from 29 healthy participants (19 females, mean age ± standard deviations (SD) = 52.4 ± 10.25 years), with no history of an outstanding neurological or psychiatric disease, were included in this study. Exclusion criteria were as follows: (i) diagnosed neurological or psychiatric disorder, (ii) a malignancy, (iii) HIV, HBV, or HCV positive, (iv) MRI-related contraindications. The study was approved by the institutional review board (IRB) at the Tel Aviv Sourasky Medical Center (TLVMC). All enrolled participants gave their informed written consent before participation.

### Test-retest reliability: Sub-study

Twelve participants (6 females, mean age ± SD = 51.5 ± 6.5 years) were followed up after 1 year and re-scanned using the same MRI protocol and the same MRI system as described below. The acquired MP-RAGE and MP2RAGE images from these participants were segmented by the different automatic segmentation methods, and the whole brain tissue, as well as DGM volumes, were assessed.

### MRI acquisition protocol

MR data were acquired using a 3 Tesla Magnetom Prisma® (Siemens, Erlangen, Germany) MR scanner, and a 20-channels phased-array head coil. The MRI protocol included a high-resolution 3D T1-weighted MP-RAGE and 3D MP2RAGE sequences. See **Table 1** for detailed

**Table 1. Acquisition parameters of magnetization prepared- rapid acquisition gradient echo (MP-RAGE) and MP2RAGE sequences.**

|  | MP-RAGE | MP2RAGE |
|---|---|---|
| Field of view (FoV) | 256×256 mm | 256×256 mm |
| Number of slices | 192 | 176 |
| Repetition time (TR) | 2200 ms | 5000 ms |
| Echo time (TE) | 3.22 ms | 3.43 ms |
| Inversion time (TI) |  |  |
| • $TI_1$ | 1100 ms | 803 ms |
| • $TI_2$ |  | 2500 ms |
| Flip angle (FA) |  |  |
| • FA 1 | 9 deg. | 4 deg. |
| • FA 2 |  | 5 deg. |
| Voxel size | 1×1×1 mm | 1×1×1 mm |
| Readout bandwidth | 150 Hz/Px | 240 Hz/Px |
| Parallel acquisition | GRAPPA, factor 2 | GRAPPA, factor 3 |
| Duration | 5:06 min | 7:07 min |

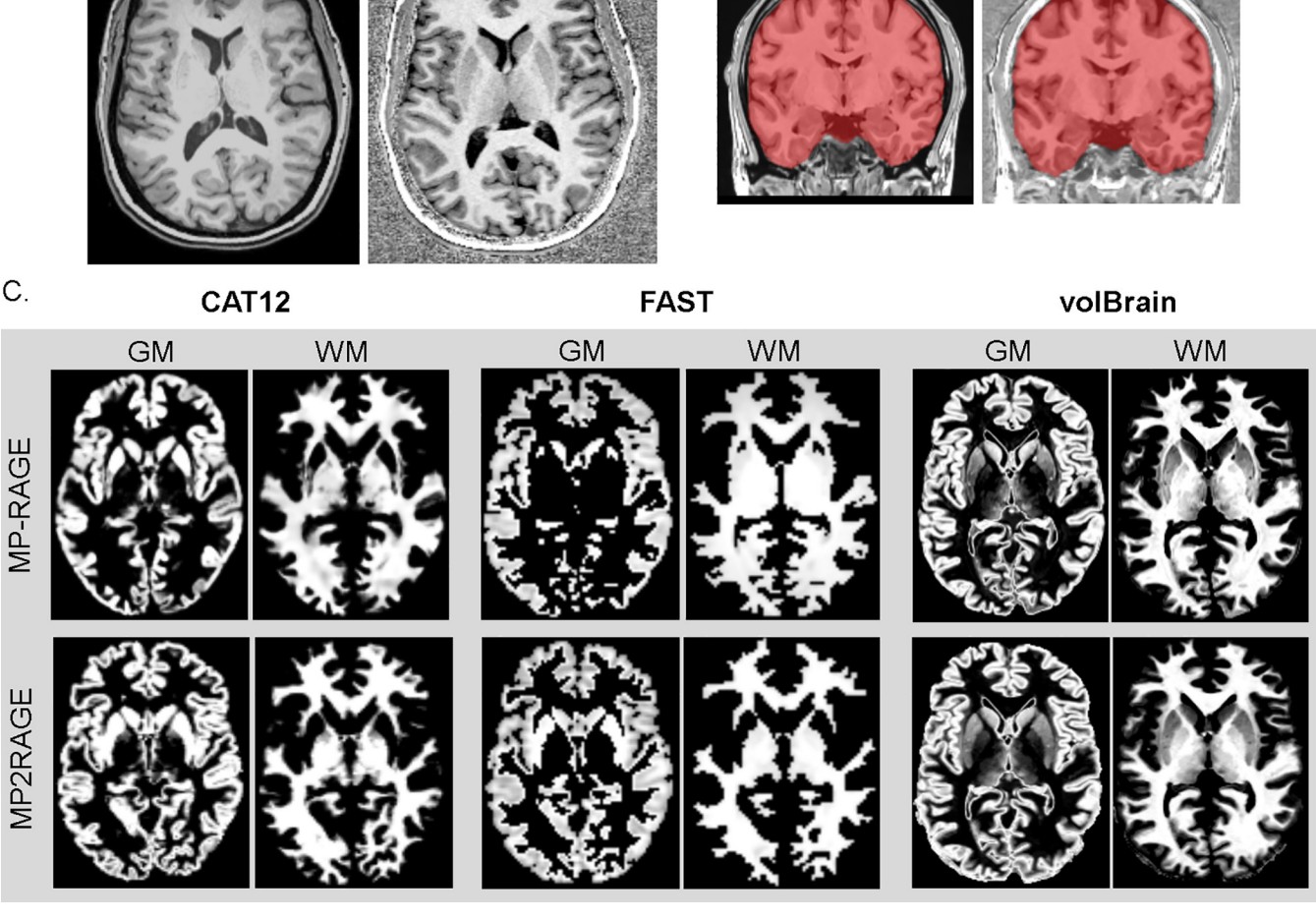

**Fig 1.** (A) Example of the acquired T1 weighted MP-RAGE and MP2RAGE images from one of the study participants. (B) Brain extraction before tissue segmentation. (C) Illustration of the tissue probability maps achieved by the used segmentation tools.

sequences parameter and **Fig 1A** for examples illustrating MRI images acquired using both sequences.

## MRI data processing

All described analyses were performed based on the acquired MP-RAGE and the MP2RAGE derived T1-weighed (UNI) images of all participants. These included CNR between GM and WM tissues ($CNR_{WG}$), whole-brain tissue, and DGM segmentation.

## Whole brain tissue segmentation

Brain tissue segmentation was performed using free and widely used automatic segmentation tools that rely on different segmentation algorithms including: (1) the computational analysis toolbox (CAT12, http://www.neuro.uni-jena.de/cat/) extension for statistical parametric mapping (SPM12, https://www.fil.ion.ucl.ac.uk/spm/software/spm12/) that uses *a-priori* tissue probability maps, to perform an initial skull-stripping, and to initialize the segmentation.

Then, it uses an adaptive maximum a posteriori segmentation approach (hypothesis-free approach) [14]. (2) volBrain processing pipeline (https://volbrain.upv.es/instructions.php) that performs non-local intracranial cavity extraction based on a library of pre-labeled brain images allowing to capture the large variability of brain shapes [15,16]. After brain extraction, mean values of the different tissue types are estimates by trimmed mean segmentation (TMS), excluding partial volume voxels from the estimation jointly with the use of an unbiased robust mean estimator [16]. (3) FMRIB's Automated Segmentation Tool (FSL-FAST, https://fsl.fmrib.ox.ac.uk/fsl/fslwiki/FAST), which is based on a hidden Markov random field model and an associated Expectation-Maximization algorithm for tissue segmentation [17] before image segmentation, bias field correction embedded in the different used pipelines and brain extraction were performed on all images. These segmentation processes yielded GM, WM, and cerebrospinal fluid (CSF) probability maps (See **Fig 1B & 1C**). The segmented probability maps achieved from the used segmentation tool were then registered to standard space (Montreal Neurological Institute, MNI) using linear image registration tool and affine transformation (12 parameters model) to achieve spatial overlap between subjects and further voxel-based morphometric comparisons (VBM).

## $CNR_{WG}$

CNR was calculated on the tissue WM and GM ($CNR_{WG}$) probability maps extracted from the acquired raw T1 images. The extracted WM and GM probability maps were thresholded at 0.9, thus reflecting only intersect areas between segmentation tools, as well as minimizing the partial volume effect. CNR between the WM and DGM ($CNR_{WdG}$) structures was defined based on the extracted WM and DGM mask FIRST (see deep grey matter segmentation). Moreover, $CNR_{WG}$ differences between both sequences were investigated sequences per lobes as well. The $CNR_{WG}$ and $CNR_{WdG}$ values were calculated as the difference between the mean values of the two tissues normalized for scan time as described in Haast et al., 2016 [18] (calculated separately for the MP-RAGE and MP2RAGE).

## Deep grey matter (DGM) segmentation

DGM segmentation was performed using: (1) FSL FIRST segmentation tool (https://fsl.fmrib.ox.ac.uk/fsl/fslwiki/FIRST), which is based on Bayesian statistical models of shape and appearance for subcortical brain segmentation [19] (2) FreeSurfer (https://surfer.nmr.mgh.harvard.edu/) that segments gray-white matter into different cortical regions based on sulci and gyri, and also segments major subcortical regions in the brain. FreeSurfer provides a full processing stream for structural MRI data, including Skull stripping, B1 bias field correction, and gray-white matter segmentation [20], and (3) volBrain labeling processing pipeline that applies novel patch-based method relying on expert manual segmentations as priors [21], producing a total of 14 DGM segments: the left and the right thalamus, caudate, putamen, pallidum, hippocampus, amygdala, and accumbens.

## Statistical analysis

VBM comparisons were performed using SPM software (SPM12; https://www.fil.ion.ucl.ac.uk/spm/software/spm12/). For the group-level voxel-wise comparisons, paired-samples t-tests were used for inter-modality comparisons (i.e, MP-RAGE vs. MP2RAGE) of whole-brain non-binary masked GM and WM tissue probability maps obtained from the different segmentation tools at baseline. Similarly, paired-samples t-tests were used to investigate the overtime changes in both tissue probability maps obtained from the different segmentation tools in the scan-rescan sub-study after 1 year. $P < 0.05$ FWE was adopted for all voxel-wise comparisons.

Using SPSS® v22 (IBM, Chicago, USA), paired-samples T-tests were used to test differences between the whole-brain tissue volumes, as well as volumes of DGM structures inter-modality comparisons. Due to the smaller sample size in the scan re-scan sub-group who underwent MRI assessment after one year, Wilcoxon's-rank test was used to test for significant differences in the whole-brain tissue volumes, as well as volumes of DGM structures. Finally, as measures for reliability, the mean percentage change, Cronbach's Alpha, as well as paired-samples T-test were calculated to explore the difference in tissue volumes assessed by the various methods based on both MRI sequences in the test-retest sub-study.

## Results

### $CNR_{WG}$

Significantly higher mean $CNR_{WG}$ was detected for the MP2RAGE vs. MP-RAGE for (Mean ±SD = 0.97 ± 0.04 and 0.8 ± 0.1 respectively; paired-samples T-test, $t_{(df = 28)} = 15.5$, $p<0.0001$). Moreover, MP2RAGE showed significantly higher $CNR_{WdG}$ compared to MP-RAGE (Mean ±SD = 0.3 ± 0.04 and 0.05 ± 0.34 respectively; paired-samples T-test, $t_{(df = 28)} = -4$, $p<0.0001$). The calculated CNR values per lobe based on MP-RAGE showed that the range of variation of CNR relative to the group mean in MPRAGE was found to be higher (9–13%) compared to MP2RAGE (1.5–1.7%).

### Whole-brain volumes

Significantly higher total brain GM volumes were obtained in MP2RAGE using the three segmentation tools (paired-samples T-test, $t_{(df = 28)} < 11$, $p<0.001$ in all cases). MP2RAGE yielded significantly higher WM total brain tissue volume in comparison to MP-RAGE when segmented using CAT-12 (paired-samples T-test, $t_{(df = 28)} > 6.4$, $p<0.001$). However, significantly lower whole-brain WM volumes were obtained from MP2RAGE using FAST and volBrain segmentation compared to MP-RAGE (paired-samples T-test, $t_{(df = 28)} = 4.9$, $p<0.001$ in both cases). Finally, in all three segmentation tools, significantly lower CSF volumes were obtained from MP2RAGE in comparison to MP-RAGE (paired-samples T-test, $t_{(df = 28)} > 3.55$, $p<0.001$ in all cases) (**Table 2** & **Fig 2**).

### Inter-modality voxel-wise differences

**Fig 3A** demonstrates the voxel-wise and whole-brain differences in tissue class classification between MP2RAGE and MP-RAGE obtained from the three segmentation tools. MP2RAGE showed significantly higher GM tissue probability in several regions compared to MP-RAGE

**Table 2. Whole-brain GM, WM, and CSF volumes obtained from the segmentation of MP-RAGE and MP2RAGE images using the different automatic segmentation tools.**

|  |  | MP-RAGE | MP2RAGE |
|---|---|---|---|
| **CAT-12** | GM** | 595.69 ± 51.2 | 622 ± 58.59 |
|  | WM** | 509 ± 45.25 | 509 ± 46.82 |
|  | CSF** | 314.4 ± 61.93 | 299.2 ± 54.44 |
| **FSL-FAST** | GM** | 585.95 ± 57.18 | 699.39 ± 67.46 |
|  | WM** | 513.06 ± 50 | 487.53 ± 45.67 |
|  | CSF** | 408.38 ± 62.67 | 336.27 ± 51.62 |
| **volBrain** | GM** | 658.9 ± 53.29 | 715.96 ± 56.98 |
|  | WM** | 524.41 ± 50.32 | 497.55 ± 52.97 |
|  | CSF** | 205 ± 50.7 | 183.22 ± 43.03 |

All values are reported in mean ± SD.

** Paired-samples T-test, $p < 0.001$. *Abbreviations; GM: Grey matter, WM: White matter, CSF: Cerebrospinal fluid.*

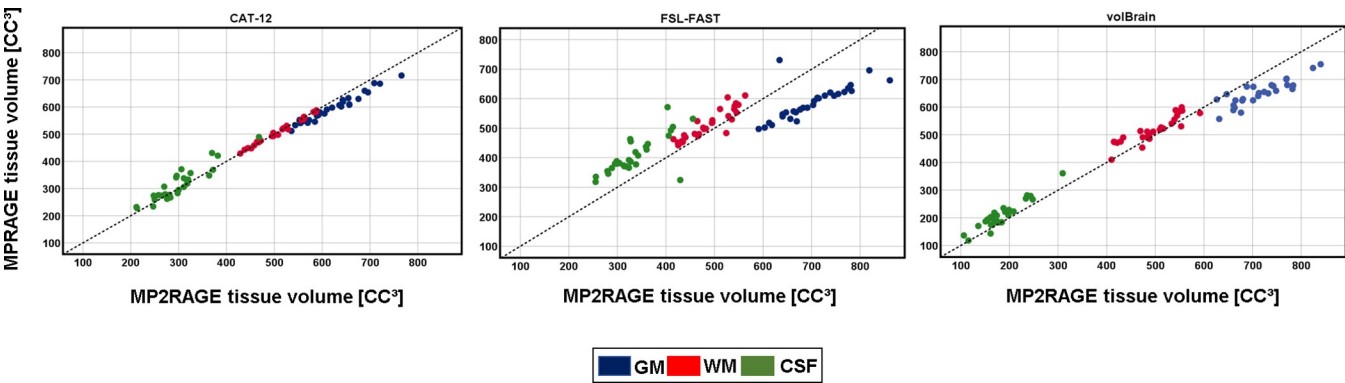

**Fig 2. Scatter plots demonstrating GM (blue), WM (red), and CSF (green) volumes measured by the different segmentation tools based on MP-RAGE and MP2RAGE sequences.** Points above the identity line (dashed) indicates higher volumes were calculated in MP-RAGE *vs.* MP2RAGE, while points below the line indicate higher values based on. MP2RAGE vs. MPRAGE.

including the bilateral thalamus, putamen, and caudate nucleus, the lingual gyrus, the inferior fronto-orbital gyri, and bilateral precentral gyrus when assessed by both CAT12 and volBrain ($p_{FWE}$<0.05). In WM, MP2RAGE showed significantly higher tissue probability in the cerebellum, frontal, temporal regions, as well as in the genu and splenium of the corpus callosum (CC) in comparison to MP-RAGE. On the other hand, MPRAGE showed higher WM densities in the bilateral thalamus, putamen and precentral gyrus (PCG) compared to MP2RAGE using CAT12 ($p_{FWE}$ < 0.05, in all cases). In FAST, MP2RAGE showed an overall significantly higher GM and WM probabilities compared to MP-RAGE. Finally, MP-RAGE showed significantly higher WM within the anterior cerebellar lobe, and bilateral putamen ($p_{FWE}$ < 0.05). No significant differences between both sequences were detected in WM when assessed by volBrain.

In the-sub study, significantly higher GM probability was found in the left thalamus in MP2RAGE vs. MP-RAGE when assessed using CAT12. In WM, MP2RAGE showed significantly higher probability in the splenium compared to MP-RAGE ($p_{FWE}$<0.05 in all cases). However, MP-RAGE showed a significantly higher WM probability in the red nucleus. No significant inter-modality differences were detected between both sequences when assessed by FAST and volBrain.

## Deep grey matter (DGM) volumes

In the three segmentation tools, MP2RAGE yielded higher mean volumes of the segmented DGM structures compared to MP-RAGE. Specifically, using FIRST, significantly higher volumes of the left putamen, left hippocampus, bilateral amygdala, accumbnes, right caudate, and right pallidum was obtained in MP2RAGE compared to MP-RAGE (paired-samples T-test, $t_{(df\ =\ 28)}$<-1.65, $p$<0.05 in all cases) (Table 3A). In volBrain, significantly higher volumes were obtained for bilateral thalamus, putamen, pallidum, the left hippocampus, and right accumbens in MP2RAGE vs. MP-RAGE (paired-samples T-test, $t_{(df\ =\ 28)}$<-2.82, $p$<0.05, in all cases). However, MP-RAGE yielded a significantly higher volume of the left caudate (paired-samples T-test, $t_{(df\ =\ 28)}$ = 3.22, $p$ = 0.003) compared to MP2RAGE (**Table 3B**). Finally, using Freesurfer, significantly higher volumes of bilateral putamen, right pallidum, and left accumbnes were obtained from MP2RAGE compared to MP-RAGE (paired-samples T-test, $t_{(df\ =\ 28)}$>4.09, $p$<0.05, in all cases). Moreover, based on MP-RAGE, Freesurfer yielded significantly higher volumes of bilateral thalamus, hippocampus, amygdala, left caudate, and left pallidum were obtained by Freesurfer compared to MP2RAGE (paired-samples T-test, $t_{(df\ =\ 28)}$< *-0.61*, $p$<0.05, in all cases) (**Table 3C**).

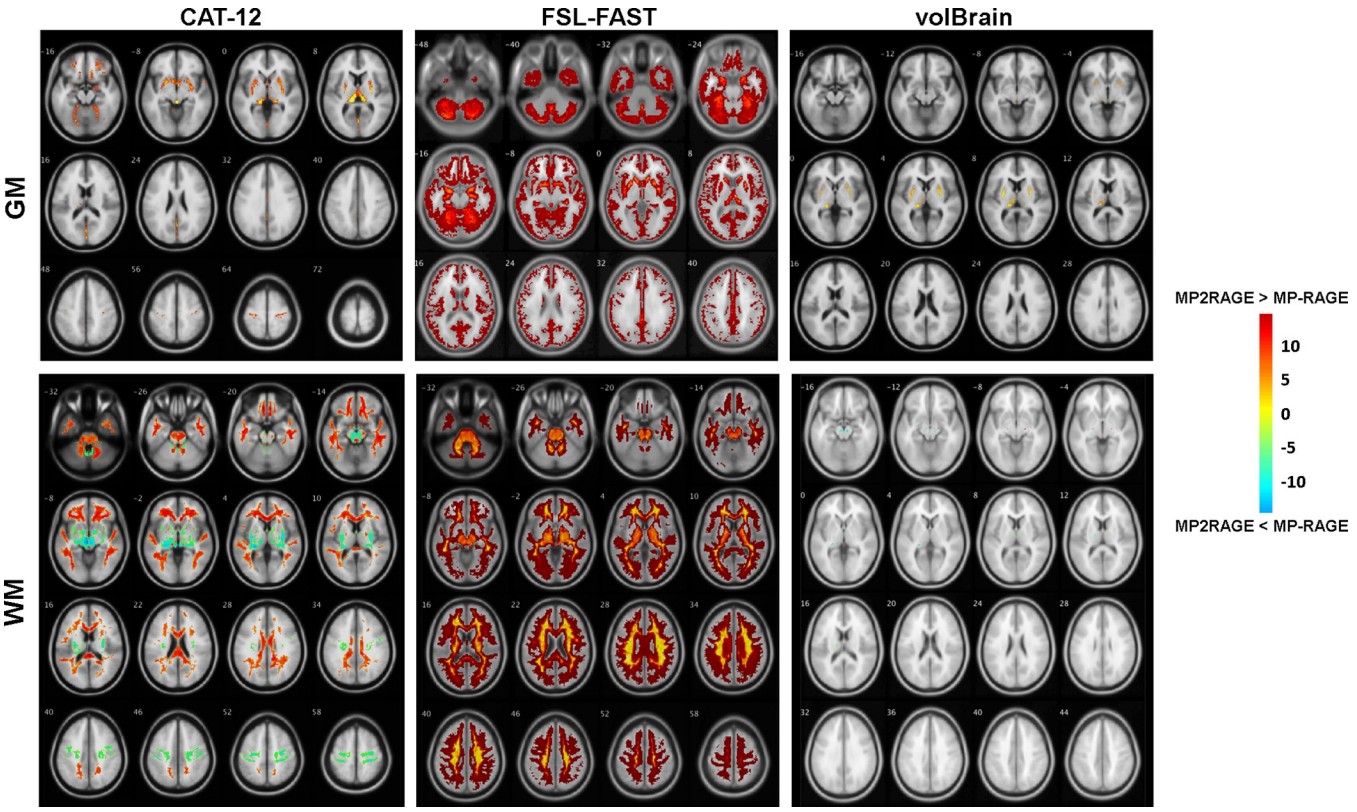

**Fig 3.** (A) Voxel-wise grey matter (GM) and white matter (WM) tissue probabilities differences when comparing MP-RAGE vs. MP2RAGE sequences based on tissue probability maps obtained by CAT12, FSL-FAST, and volBrain ($p_{FWE} < 0.05$).

### Test-retest reliability: Sub-study

At re-scan, MP2RAGE demonstrated higher whole-brain GM, lower WM, and CSF volumes compared to MP-RAGE in all three used tools. In the used automatic segmentation methods a higher mean percentage of change was observed in total brain GM assessed based on MP2RAGE compared to MP-RAGE. On the other hand, MP-RAGE demonstrated a higher overtime percentage of change in WM and DGM volumes compared to MP2RAGE. Across all used segmentation methods, high scan-rescan reliability was obtained in both sequences as reflected by Cronbach's α coefficients (**Table 4**). No significant differences were observed in the obtained whole-brain and DGM volumes based on both sequences from all used tools ($p > 0.05$; in all cases). In VBM, significantly higher GM probability was found in the left thalamus in MP2RAGE vs. MP-RAGE when assessed using CAT12). In WM, MP2RAGE showed a significantly higher probability in the splenium compared to MP-RAGE ($p_{FWE} < 0.05$). Additionally, MP-RAGE showed significantly higher WM probability in the red nucleus. No significant inter-modality differences were detected between both sequences when assessed by FAST and volBrain (**Fig 4**).

### Discussion

We compared the results of brain tissue segmentation performed on two MRI sequences (MP2RAGE vs. MP-RAGE) using different automatic segmentation tools. Based on the obtained results, MP2RAGE yielded higher $CNR_{WG}$ $CNR_{WdG}$, and consequently, resulted in higher whole-brain GM and DGM, as well as lower CSF volumes when compared to MP-RAGE.

**Table 3. Measured deep grey matter (DGM) structures volumes using the different segmentation tools based on MP-RAGE and MP2RAGE images.**

| Method | DGM structure | MP-RAGE | MP2RAGE | *p*- value |
|---|---|---|---|---|
| **A. FIRST** | L Thalamus | 7.87 ± 0.66 | 7.9 ± 0.64 | *n.s* |
| | L Caudate | 3.40 ± 0.4 | 3.38 ± 0.4 | *n.s* |
| | L Putamen | 4.84 ± 0.5 | 5.0 ± 0.54 | ≤ 0.001 |
| | L Pallidum | 1.76 ± 0.26 | 1.8 ± 0.24 | *n.s* |
| | L Hippocampus | 3.66 ± 0.46 | 3.9 ± 0.55 | ≤ 0.001 |
| | L Amygdala | 1.21 ± 0.27 | 1.3 ± 0.27 | ≤ 0.01 |
| | L Accumbens | 0.5 ± 0.12 | 0.55 ± 0.1 | ≤ 0.001 |
| | R Thalamus | 7.72 ± 0.71 | 7.7 ± 0.74 | *n.s* |
| | R Caudate | 3.41 ± 0.4 | 3.56 ± 0.41 | ≤ 0.05 |
| | R Putamen | 5.0 ± 0.55 | 4.94 ± 0.59 | *n.s* |
| | R Pallidum | 1.82 ± 0.23 | 1.87 ± 0.23 | ≤ 0.05 |
| | R Hippocampus | 3.72 ± 0.49 | 3.96 ± 0.55 | *n.s* |
| | R Amygdala | 1.28 ± 0.22 | 1.51 ± 0.22 | ≤ 0.001 |
| | R Accumbens | 0.41 ± 0.1 | 0.45 ± 0.08 | ≤ 0.01 |
| **B. volBrain** | L Thalamus | 5.58 ± 0.47 | 5.68 ± 0.39 | ≤ 0.001 |
| | L Caudate | 3.41 ± 0.41 | 3.4 ± 0.42 | ≤ 0.01 |
| | L Putamen | 4.12 ± 0.43 | 4.17 ± 3.6 | ≤ 0.001 |
| | L Pallidum | 1.16 ± 0.13 | 1.26 ± 0.14 | ≤ 0.001 |
| | L Hippocampus | 3.76 ± 0.45 | 3.73 ± 0.4 | ≤ 0.01 |
| | L Amygdala | 0.8 ± 0.11 | 0.8 ± 0.12 | *n.s* |
| | L Accumbens | 0.33 ± 0.06 | 0.33 ± 0.05 | *n.s* |
| | R Thalamus | 5.56 ± 0.5 | 5.61 ± 0.42 | ≤ 0.001 |
| | R Caudate | 3.41 ± 0.41 | 4.44 ± 5.26 | *n.s* |
| | R Putamen | 4.1 ± 0.45 | 4.1 ± 0.34 | ≤ 0.001 |
| | R Pallidum | 1.18 ± 0.13 | 1.27 ± 0.14 | ≤ 0.001 |
| | R Hippocampus | 3.73 ± 0.66 | 3.71 ± 0.77 | *n.s* |
| | R Amygdala | 0.81 ± 0.12 | 0.81 ± 0.12 | *n.s* |
| | R Accumbens | 0.3 ± 0.06 | 0.31 ± 0.05 | ≤ 0.01 |
| **C. FreeSurfer** | L Thalamus | 7.09 ± 0.7 | 6.73 ± 0.67 | ≤ 0.001 |
| | L Caudate | 3.52 ± 0.44 | 3.34 ± 0.4 | ≤ 0.001 |
| | L Putamen | 5.29 ± 0.53 | 5.43 ± 0.66 | ≤ 0.01 |
| | L Pallidum | 1.66 ± 0.17 | 1.8 ± 0.22 | ≤ 0.001 |
| | L Hippocampus | 4.17 ± 0.44 | 3.84 ± 0.44 | ≤ 0.001 |
| | L Amygdala | 1.57 ± 0.22 | 1.43 ± 0.24 | ≤ 0.001 |
| | L Accumbens | 0.51 ± 0.1 | 0.55 ± 0.1 | ≤ 0.05 |
| | R Thalamus | 7.24 ± 0.74 | 6.68± 0.64 | ≤ 0.001 |
| | R Caudate | 3.43 ± 0.44 | 3.44 ± 0.42 | *n.s* |
| | R Putamen | 5.0 ± 0.55 | 5.21 ± 0.58 | ≤ 0.001 |
| | R Pallidum | 1.49 ± 0.16 | 1.64 ± 0.16 | ≤ 0.001 |
| | R Hippocampus | 4.31 ± 0.42 | 4.02 ± 0.41 | ≤ 0.001 |
| | R Amygdala | 1.6 ± 0.2 | 1.5 ± .21 | ≤ 0.001 |
| | R Accumbens | 0.58 ± 0.09 | 0.59 ± 0.08 | *n.s* |

All values reported in mean ± SD. Abbreviations: L: Left, R: Right.

Compared to MP-RAGE, significantly higher CNR in whole-brain and DGM (80% & 16% respectively), as well as lower within-group variance were calculated in MP2RAGE images based on the segmented tissue probability maps. This can be explained by the differences

**Table 4. Test-re-test of the mean percentage of overtime change and Cronbach's *α* in whole-brain GM and WM, as well as deep grey matter (DGM) volumes assessed using the different automatic tools based on MP-RAGE and MP2RAGE.**

| | | | MP-RAGE | MP2RAGE |
|---|---|---|---|---|
| **Whole brain** | CAT-12 | GM | 1.17±0.33/0.99 | 2.3±0.76/0.98 |
| | | WM | 0.78±0.18/0.99 | 1.14±0.30/0.99 |
| | FSL-FAST | GM | 4.02±1.18/0.92 | 3.56±0.91/0.97 |
| | | WM | 6.07±1.92/0.90 | 7.14±1.96/0.90 |
| | volBrain | GM | 1.51±0.38/0.99 | 2.1±0.65/0.94 |
| | | WM | 1.7±0.29/0.99 | 6.18±1.92/0.92 |
| **DGM** | FSL-FIRST | | 5.08±2.30/0.96 | 3.32±1.76/0.96 |
| | Freesurfer | | 4.44±2.28/0.94 | 4.13±1.90/0.96 |
| | volBrain | | 1.03±2.53/0.94 | 3.20±3.76/0.93 |

All values are reported in mean ± SE/ Cronbach's α.

between the two sequences, and the fact that increased rates of bias-field effects are inherent in MP-RAGE images at high and ultra-high magnetic fields (≥3 Tesla), whereas, MP2RAGE is less sensitive to B1 bias, enabling the acquisition of images with enhanced CNR [3,9,14]. Additionally, MP2RAGE uses a higher bandwidth compared to MPRAGE (240 Hz/Px vs. 150 Hz/Px). This use of high bandwidth is expected to reduce susceptibility effects including eddy currents associated with metal, and thus improve image SNR [3].

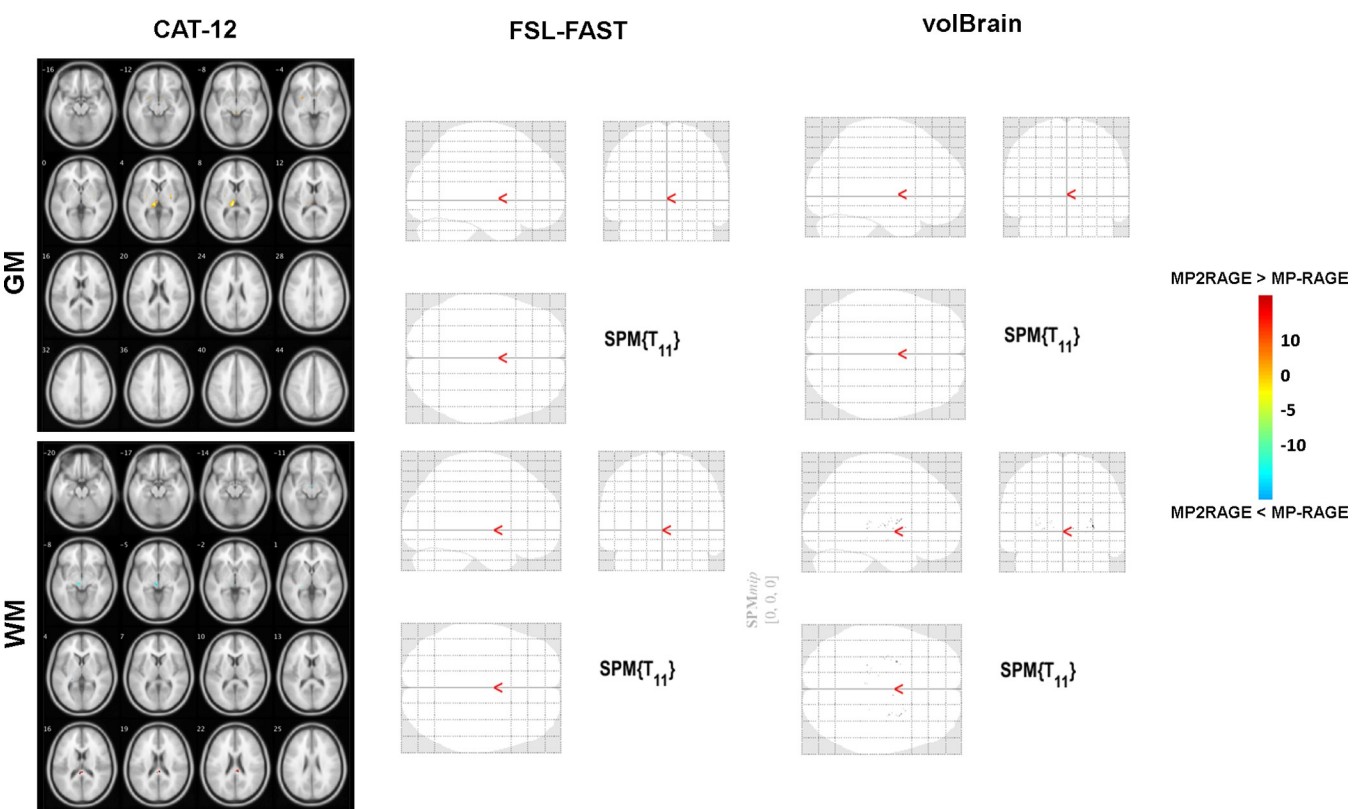

**Fig 4. Sub-study VBM results of inter-modality differences in grey matter (GM) and white matter (WM) tissue probabilities differences based on tissue probability maps obtained by CAT12, FSL-FAST, and volBrain.**

More accurate classification of tissue boundaries especially in GM and DGM due to the enhanced image quality of MP2RAGE images was observed, especially using tools that rely on edge detection such as CAT-12 and volBrain. This was evident from the voxel-wise comparisons where higher GM probabilities in MP2RAGE were observed mainly in several ventral brain regions, including the thalamus and putamen, as well as cortical sulci, all of which suffer low-degree of tissue boundary detection accuracy [12,22,23]. This finding was further validated by the calculated whole-brain volumes, where significantly higher whole-brain GM and lower CSF volumes were obtained based on MP2RAGE images compared to MP-RAGE. Image quality on an image plays a central role in the segmentation and quantitative assessment of the brains' tissue [3,24]. Yet, unified MP2RAGE images contain a noisy background, which might affect the accuracy of tissue segmentation [25,26]. This can be overcome using brain extraction and skull stripping algorithms that can be applied prior to segmentation [8,27].

CAT12 and volBrain yielded comparable results in the voxel-based as well as in whole-brain tissue volume comparisons in the cross-sectional VBM analysis. Both of these approaches apply prior spatial information for each tissue class based on the standard MNI template [21,28,29]. Whereas by relying on signal intensity within each voxel, FSL-FAST applies a discrete Markov random field (MRF) to obtain prior spatial information for each voxel, and then iteratively updates its assignment to the correct tissue class [17]. This explains the obtained inter-modality voxel-wise differences using FAST segmentation, where due to its' higher CNR, higher GM probabilities were observed in MP2RAGE, suggesting that FAST might be advantageous for segmenting low-quality images, as it is designed to be robust against lower signal-to-noise ratio (SNRs) [14,17]. Meanwhile, due to the enhanced CNR of MP2RAGE, automatic segmentation tools that rely on boundary detection as the basis for tissue classification would yield a more accurate estimation of volumetric and thickness measures [30,31]. This is further implicated by the findings of the sub-study inter-modality VBM analysis, in which significant GM and WM differences between both sequences were observed only in CAT12, illustrating the robustness of the segmentation algorithms implemented in FAST and volBrain compared to CAT12.

Automatic DGM segmentation tools often yield inconsistent results as these brain structures have shorter T1 values and decreased contrast between subcortical GM and the neighboring WM structures [32]. Considering that the used segmentation methods were developed and applied mainly for the processing of T1-weighted images (MP-RAGE, SPGR), comparable DGM volumes were obtained from MP-RAGE using the three tools. However, by faster evaluation of T1 values [3,33], MP2RAGE overcomes T1 decay and consequently resulting in better CNR in these regions [9]. Indeed, higher tissue probabilities of the putamen, caudate, and thalamus boundaries were observed using CAT12 and volBrain based on the voxel-wise comparisons. Moreover, significantly higher volumes were obtained based on MP2RAGE compared to MP-RAGE, mainly when using FIRST and volBrain. This is in line with previously reported findings by Okubo et al., where higher DGM probabilities were detected in MP2RAGE vs. MP-RAGE in several structures including the putamen, caudate, thalamus, and substantia nigra [9]. Conversely, using Freesurfer, higher volumes in several structures such as the hippocampus and the amygdala were obtained with MP-RAGE compared to MP2RAGE. This could be attributed to the poor delineation of these subcortical structures due to its basal location in the brain and the reduced SNR in MP-RAGE as a result [12]. Such observed discrepancies further emphasize that pooling together and interpreting the findings of the various studies, where different segmentation tools have been implemented can be limited and should be taken with caution [32].

Due to the lack of a "gold standard', we performed a test-retest analysis based on a subgroup of participants who were scanned after one year. Typically, manual segmentation is

performed by an experienced operator. However, applying this method for both whole-brain and DGM segmentation is labor-intensive and is prone to tracer bias [23]. Based on the substudy, varying mean percentage of change and within-group variance in the measured whole brain, as well as DGM structures, were obtained for MP2RAGE vs. MP-RAGE. MP2RAGE demonstrated a higher mean percentage of change in whole brain volume, and lower in DGM volumes compared to MP-RAGE. The observed higher percentage of annual whole-brain volume change based on MP2RAGE falls in the range of normative values reported by a longitudinal study in normal aging [34]. Since our studied subjects were above the age of 50, these findings strongly suggest that overcoming partial volume effects that are inherent in weighted images [35], MP2RAGE might be more sensitive to subtle overtime changes in tissue volumes with age. In clinical trials, this would lead to enhanced statistical power, thus allowing for the detection of possible effects of potential drugs or treatments.

The limitations of the study are: firstly, this was a single-site study. Further studies enrolling larger sample sizes in multi-centric settings would contribute to better establish the reproducibility of volumetric measurements based on MP2RAGE. Additionally, we systematically demonstrate differences in whole-brain GM, WM, and DGM assessed by the different segmentation tools. Further e inter-and-intra modality investigations are needed to determine differences in cortical thickness measures and to establish whether the net gain of MP2RAGE would out-weight the cost of the additional acquisition time.

In conclusion, the obtained results indicate that MP2RAGE offers enhanced CNR compared to MP-RAGE. Using the same segmentation tools for multi-center or longitudinal studies is essential for more reproducible tissue classification and optimal segmentation results. Therefore, it is a strong candidate to be used for whole-brain and DGM segmentation in future basic and clinical trials, allowing more sensitive structural metrics for the monitoring of the progression of many diseases and outcomes of the different treatment strategies.

## Author Contributions

**Conceptualization:** Amgad Droby, Dafna Ben Bashat, Moran Artzi.

**Data curation:** Nir Giladi, Anat Mirelman.

**Formal analysis:** Amgad Droby, Moran Artzi.

**Funding acquisition:** Nir Giladi, Anat Mirelman.

**Investigation:** Avner Thaler, Anat Mirelman.

**Methodology:** Amgad Droby, Dafna Ben Bashat, Moran Artzi.

**Project administration:** Nir Giladi, Anat Mirelman.

**Resources:** Anat Mirelman.

**Supervision:** Anat Mirelman, Moran Artzi.

**Writing – original draft:** Amgad Droby, Moran Artzi.

**Writing – review & editing:** Amgad Droby, Avner Thaler, Nir Giladi, R. Matthew Hutchison, Anat Mirelman, Dafna Ben Bashat, Moran Artzi.

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
