## [Decision Letter · Decision Letter 0]

3 Feb 2021

Pécs, Hungary

February 2, 2021

PONE-D-20-40185

Whole brain and deep gray matter structure segmentation: quantitative comparison between MPRAGE and MP2RAGE sequences

PLOS ONE

Dear Dr. Droby,

Thank you for submitting your manuscript to PLOS ONE. After careful consideration, we feel that it has merit but does not fully meet PLOS ONE’s publication criteria as it currently stands. Therefore, we invite you to submit a revised version of the manuscript that addresses the points raised by the Reviewers, listed below.

We look forward to receiving your revised manuscript.

Kind regards,

Joseph Najbauer, Ph.D.

Academic Editor

PLOS ONE

Journal Requirements:

2. Thank you for stating the following in the Competing Interest section:

"NG: serves as a member of the Editorial Board for the Journal of Parkinson's Disease. He serves as consultant to Sionara, Accelmed, Teva, NeuroDerm, Intec Pharma, Pharma2B, Denali and Abbvie. He receives royalties from Lysosomal Therapeutics (LTI) and payment for lectures at Teva, UCB, Abbvie, Sanofi-Genzyme, Bial and Movement Disorder Society. He received research support from the Michael J Fox Foundation, the National Parkinson Foundation, the European Union 7th Framework Program and the Israel Science Foundation as well as from Teva NNE program, Biogen, LTI, and Pfizer. RMH is an employee of and owns stock in Biogen. JC: is a former employee of Biogen. "

We note that one or more of the authors have an affiliation to the commercial funders of this research study : Biogen Inc.

2.1. Please provide an amended Funding Statement declaring this commercial affiliation, as well as a statement regarding the Role of Funders in your study. If the funding organization did not play a role in the study design, data collection and analysis, decision to publish, or preparation of the manuscript and only provided financial support in the form of authors' salaries and/or research materials, please review your statements relating to the author contributions, and ensure you have specifically and accurately indicated the role(s) that these authors had in your study. You can update author roles in the Author Contributions section of the online submission form.

2.2. Please also provide an updated Competing Interests Statement declaring this commercial affiliation along with any other relevant declarations relating to employment, consultancy, patents, products in development, or marketed products, etc.  

Reviewers' comments:

Reviewer's Responses to Questions

**Comments to the Author**

1. Is the manuscript technically sound, and do the data support the conclusions?

Reviewer #1: Yes

Reviewer #2: Partly

2. Has the statistical analysis been performed appropriately and rigorously? 

Reviewer #1: Yes

Reviewer #2: I Don't Know

3. Have the authors made all data underlying the findings in their manuscript fully available?

Reviewer #1: No

Reviewer #2: No

4. Is the manuscript presented in an intelligible fashion and written in standard English?

Reviewer #1: Yes

Reviewer #2: Yes

5. Review Comments to the Author

Reviewer #1: The authors investigated the agreement of brain tissues (white matter, grey matter and deep grey matter) volumes obtained from MP2RAGE compared to MPRAGE. The authors compared the volumes obtained from different segmentation methods at baseline and 1 year follow-up to assess the accuracy and the reproducibility of volumetric measures.

The authors showed the relevance of using MP2RAGE to estimate brain volumes.

I have few comments and questions to the authors:

Material and Methods

-Please provide more detailed description of the used segmentation methods.

-Please provide demographical information for the longitudinal group.

-Please explain why you did not used a removing background noise method on the MP2RAGE and N3/N4 bias field correction on the MPRAGE before tissues classification (O’Brian et al., 2014) ?

Results

-Please provide the CNR difference between DGM and WM.

-Please provide a Figure for each tissue segmentation estimated by the used methods.

-Please provide the results of the intra-modality voxel-wise differences between baseline and follow-up as well as the inter-modality voxel-wise differences at follow-up.

-What about inter and intra-modality cortical thickness differences at baseline and follow-up?

-Table 4: please provide mean percentage change overtime for WM volume.

Discussion

-Please discuss the potential effect of background noise on tissues classification.

-Please update your discussion according to my comments/questions.

Reviewer #2: In this work, Droby et al. compared whole brain, cortical and subcortical gray, and white matter tissue segmentations between results based on MP2RAGE and MPRAGE data and different segmentation software, as well as between data acquired at different time points (1 yr apart). Across a total of 29 subjects, they found that MP2RAGE data is characterized by significant higher CNR leading to differences in subcortical and white and gray matter tissue probabilities, depending on the segmentation tool used. Comparative studies such as this are important to validate popular sequences, in this case MPRAGE and MP2RAGE, for anatomical imaging. Especially clinical researchers that rely on accurate segmentation of brain structures will benefit from this work and/or enable them to better design acquisition and analysis protocols. Therefore, I see the value of this manuscript but think it must take care of the following major issues to consider publication:

First, with regards to the impact of data quality and CNR analysis:

1. The authors define the sequence-specific CNR as the difference between the mean values of GM and WM, divided by the sum of squares of their standard deviation. However, ideally, CNR values need to be corrected for differences in total acquisition time (T) as longer acquisition (such as for the MP2RAGE sequence) positively affect achievable CNR. As such, current CNR needs to be converted to CNR per unit time, by normalizing it to the square root of T (in minutes) as for example done in https://doi.org/10.3389/fnana.2016.00112. This, as well as the fact that the MP2RAGE T1w image is a combination of two separate inversion images, and thus based on more data, needs to be further discussed to put the results in perspective.

2. Considering that the quality of the data might vary across the brain due to B1 transmit and receive homogeneity differences, as mentioned in the introduction by the authors too, I think it would be of interest to see whether difference in CNR between MPRAGE and MP2RAGE varies across the brain, and whether this spatial variation is more prominent for MPRAGE or MP2RAGE. This could be tested by averaging per lobe for example and will also ease interpretation of the observed differences between tissue probability maps.

3. In line my previous comment and with the authors’ hypothesis stated in the discussion that the observed differences in CNR “can be explained by the differences between the two sequences, and the fact that increased rates of bias-field effects are inherent in MP-RAGE images at high and ultra-high magnetic fields (≥3 Tesla), whereas, MP2RAGE is less sensitive to B1 bias, thus enabling the acquisition of images with enhanced” I would propose to calculate subject-specific difference maps between the MP-RAGE and MP2RAGE images (e.g., after normalization to the average CSF value, for example). This will provide a surrogate bias map and might provide additional information with regards to origin of the observed differences in terms of tissue segmentation. See also https://doi.org/10.3389/fnana.2016.00112 and https://doi.org/10.1016/j.neuroimage.2020.117373 for relevant demonstrations and discussions on the effect of B1 biases on cortical and subcortical segmentations using MP2RAGE data.

With regards to the tissue probability map comparisons:

4. Import details are lacking with regards to the voxel-based analyses of the tissue probability maps. The authors should elaborate more on the SPM analyses as it is impossible now for the reader to replicate the current analyses. For example, which registration tool was used? And why was chosen for a linear registration instead of non-linear transformation? The latter will ensure larger overlap across subjects for statistical analyses and more power. Moreover, did the authors coregister each MP-RAGE and MP2RAGE image individually to the template space or did they use the same transformation matrix for both types of images? This coregistration process should be clarified to rule out potential sources of biases in the comparison.

5. Finally, it is confusing why the WM results using volBrain are not shown in Figure 2 as for CAT-12 and FSL-FAST and as suggested in the caption.

In general:

6. Especially in the introduction the authors should better emphasize the novelty of the current work (e.g., compared to https://doi.org/10.1002/jmri.24960).

7. I think the paper would benefit from an extra table/figure that summarizes all findings. This will make it easier for the reader to identify the key message of the manuscript and follow the narrative of the discussion.

Minor points

Introduction:

8. Define ‘high’ and ‘ultra-high’ magnetic field strengths

9. Last paragraph, ‘doesn’t’ to ‘does not’

Test-retest reliability (methods):

10. Add characteristics for the test-retest subjects

11. First sentence: ‘described above’ to ‘described below’ or move paragraph

12. Second sentence:

a. Change ‘T1-MPRAGE’ to MP-RAGE to follow previous notations

b. Change ‘patients’ to ‘participants’

MRI acquisition protocol (methods):

13. No need to define MP-RAGE again.

14. Remove sequence parameters from text as this is redundant with Table 1. Also add readout bandwidth to better estimate SNR and interpret the CNR differences.

15. Last sentence: would be informative for the reader to add a bit more detail what the authors are referring to with ‘Table 1 & Figure 1’

MRI data processing (methods):

16. As for MP-RAGE, no need to define CNR again.

Whole brain tissue segmentation (methods):

17. First sentence: change ‘free-widely’ to ‘free and widely’

18. Last sentence:

c. ‘Montreal neurological institute’ to ‘Montreal Neurological Institute’

Contrast-to-noise (methods):

19. Subscript ‘WG’ in ‘CNRWG’ as used in the equation and later in the results

DGM segmentation (methods):

20. I do not understand why there is a difference in the tools used for whole-brain and DGM segmentation. I would suggest using the same set of tools across both analyses or motivate why it was chosen not to.

Statistical analysis (methods)s:

21. Is there are specific reason why the authors did not use a multivariate (i.e., across multiple regions of interest) ANOVA test for the statistical differences? This will also allow to test for the effect of segmentation tools, region of interest and/or potential interactions.

Whole-brain volumes (results):

22. I think it would be helpful for the reader to see example tissue segmentation results for a single subject using both types of data. This could for example be combined with Figure 1.

DGM segmentation (results):

23. Second sentence: ‘accumbnes’ to ‘accumbens’

Test-retest reliability (results):

24. First sentence:

d. ‘MP2RGAE’ to ‘MP2RAGE’

e. Higher what? Volume? Please specify

6. PLOS authors have the option to publish the peer review history of their article (what does this mean?). If published, this will include your full peer review and any attached files.

Reviewer #1: No

Reviewer #2: No

---

## [Author Response · Author response to Decision Letter 0]

6 Apr 2021

Response to reviewers' comments 

We would like to thank the esteemed reviewers deeply for their positive evaluation of our manuscript, and their valuable feedback. Below, are our point-by-point response to the various comments.

Reviewer #1

The authors investigated the agreement of brain tissues (white matter, grey matter and deep grey matter) volumes obtained from MP2RAGE compared to MPRAGE. The authors compared the volumes obtained from different segmentation methods at baseline and 1 year follow-up to assess the accuracy and the reproducibility of volumetric measures.

The authors showed the relevance of using MP2RAGE to estimate brain volumes.

I have few comments and questions to the authors:

Material and Methods

1. Please provide more detailed description of the used segmentation methods.

Response: We agree with the reviewer that a detailed description of the applied segmentation tools would be helpful for the readers. We have modified the corresponding sections in the methods adding the following information:

1. Brain tissue segmentation: was performed using widely used automatic segmentation tools that rely on different segmentation algorithms including: (1) the computational analysis toolbox (CAT12, http://www.neuro.uni-jena.de/cat/) extension for statistical parametric mapping (SPM12, https://www.fil.ion.ucl.ac.uk/spm/software/spm12/) that uses a-priori tissue probability maps, to perform an initial skull-stripping, and to initialize the segmentation. Then, it uses an adaptive maximum a posteriori segmentation approach (hypothesis-free approach) [Rajapakse et al., 1997]. (2) volBrain processing pipeline (https://volbrain.upv.es/instructions.php) that performs non-local intracranial cavity extraction based on a library of prelabeled brain images allowing to capture the large variability of brain shapes [Manjon et al., 2014; Majon et al., 2016]. After brain extraction, mean values of the different tissues types are estimates by trimmed mean segmentation (TMS), excluding partial volume voxels from the estimation jointly with the use of an unbiased robust mean estimator [Manjón et al., 2008; Manjon et al., 2016]. (3) FMRIB's Automated Segmentation Tool (FSL-FAST, https://fsl.fmrib.ox.ac.uk/fsl/fslwiki/FAST), which is based on a hidden Markov random field model and an associated Expectation-Maximization algorithm for tissue segmentation [Zhang et al., 2001] Brain extraction (BET) was performed prior to image segmentation. 

2. Deep grey matter segmentation: the following tools were used: DGM segmentation was performed using: (1) FSL FIRST segmentation tool (https://fsl.fmrib.ox.ac.uk/fsl/fslwiki/FIRST), which is based on bayesian statistical models of shape and appearance for subcortical brain segmentation [Patenuade et al., 2011] (2) FreeSurfer (https://surfer.nmr.mgh.harvard.edu/) that segments gray-white matter into different cortical regions based on sulci and gyri, and also segments major subcortical regions in the brain. FreeSurfer provides a full processing stream for structural MRI data, including Skull stripping, B1 bias field correction, and gray-white matter segmentation [Fischl et al., 2002], and (3) volBrain labeling processing pipeline that applies novel patch-based method relying on expert manual segmentations as priors [Coupe et al., 2011].

2. Please provide demographical information for the longitudinal group.

Response: In the longitudinal group, N=12 participants (6 females, mean age ± SD = 51.5 ± 6.5 years) were included, and re-scanned after 1 year using the same MRI system and protocol. This information was now added to the corresponding section in the Methods.

3. Please explain why you did not used a removing background noise method on the MP2RAGE and N3/N4 bias field correction on the MPRAGE before tissues classification (O’Brian et al., 2014) ?

Response: We apologize that we failed to provide this information in the methods section. In fact, all images underwent brain extraction prior to the segmentation process results including bias field correction, skull, and background noise removal using the pipelines embedded within the used pipelines (i.e, CAT12, FAST, FIRST, Freesurfer, and volBrain). We added this information in the corresponding section of the methods.

Results

4. Please provide the CNR difference between DGM and WM. 

Response: CNR between the WM and DGM (CNRWdG) structures were calculated based on the extracted WM DGM masks obtained by FIRST. Significantly higher mean CNRWdG was detected for the MP2RAGE vs. MP-RAGE (Mean±SD = 0.98±0.13 and 0.16±1.00 respectively; p<0.0001). This information was added to the corresponding sections in the Methods and Results.

5. Please provide a Figure for each tissue segmentation estimated by the used methods.

Response: Thank you for this suggestion. We have included a figure (Figure 1B) illustrating the tissue probability maps achieved by the different segmentation tools.

6. Please provide the results of the intra-modality voxel-wise differences between baseline and follow-up as well as the inter-modality voxel-wise differences at follow-up.

Response: At pFWE<0.05, significant higher GM probability were found in the left thalamus in MP2RAGE vs. MP-RAGE when assessed using CAT12. In WM, MP2RAGE showed significantly higher probability in the splenium compared to MP-RAGE. However, MP-RAGE showed significantly higher WM probability in the red nucleus. No significant inter-modality differences were detected between both sequences when assessed by FAST and volBrain. This finding might be indicative of robustness of the segmentation algorithms implemented in FAST and volBrain compared to CAT12, an in line with the values in Table 4, where CAT12 showed the highest percentage of change over time compared to FAST and volBrain. Moreover, in the sub-study of 12 participants, no significant intra-modality differences were observed between both time points in all three used automatic tools. We believe that these differences are possibly due to the relatively small group, and hence low statistical power. We added these findings to the corresponding section of the results and Figure 4.

7. What about inter and intra-modality cortical thickness differences at baseline and follow-up?

Response: We thank the reviewer for this comment. We opted to not include inter-and intra-modality differences in regards to cortical thickness measures manuscript due to the following considerations. Firstly, among the investigated and used segmentation pipelines, only Freesurfer and CAT12 offer cortical thickness measures, while no such measures can be obtained by the other tested pipelines (i.e, volBrain, and FAST). More importantly, both CAT12 and Freesurfer rely on different atlases for cortical parcellation and regional labeling (CAT 12 uses Neuromorphometrics [http://neuromorphometrics.com], while Fressurfer relies on Desikan-Killiany Atlas/ Destrieux Atlas]. In this case, it would be inaccurate to infer about inter-and- intra modality comparison of the thickness assessed using both of these tools at baseline [See Tustison et al., 2014]. Therefore, we saw that it would be more meaningful to focus on global volumetric measures in order maintain coherence throughout the manuscript, especially since the accuracy of thickness estimation heavily depends on the results of tissue segmentation step [Clarkson et al., 2011; Neuroimage]. To this end, we have highlighted this as a limitation of this study the current study. We do nevertheless agree with the reviewer that this issue would be of high relevance for future follow-up studies that could address the inter-and-intra modality differences of cortical thickness measures achieved from various tools in a more systematic and coherent fashion. 

8. Table 4: please provide mean percentage change overtime for WM volume.

Response: Table 4 was now updated. We inserted mean percentage of overtime change for both GM and WM.

Discussion

9. Please discuss the potential effect of background noise on tissues classification.

Response: Background noise in the image can affect the accuracy of tissue classification, and consequently, on the measured volume. However and brain extraction tools show to overcome this and improve tissue segmentation [Smith, 2002; Tustison et al., 2010; O'Brien et al., 2014]. This is of importance in case of MP2RAGE, where the performance of the different automatic segmentation tools maybe challenged due to the noisy background of the UNI images. As now detailed in the methods section, in order to overcome this problem, brain extraction step was carried out on all segmented images using the different applied tools. Moreover, we added an illustration of the step in Figure 1B. As suggested by the reviewer, this point was also addressed in the discussion as well.

10. Please update your discussion according to my comments/questions.

Response: The discussion was updated based on the new additions based on comments 4, 6-9. 

Reviewer #2

In this work, Droby et al. compared whole brain, cortical and subcortical gray, and white matter tissue segmentations between results based on MP2RAGE and MPRAGE data and different segmentation software, as well as between data acquired at different time points (1 yr apart). Across a total of 29 subjects, they found that MP2RAGE data is characterized by significant higher CNR leading to differences in subcortical and white and gray matter tissue probabilities, depending on the segmentation tool used. Comparative studies such as this are important to validate popular sequences, in this case MPRAGE and MP2RAGE, for anatomical imaging. Especially clinical researchers that rely on accurate segmentation of brain structures will benefit from this work and/or enable them to better design acquisition and analysis protocols. Therefore, I see the value of this manuscript but think it must take care of the following major issues to consider publication:

First, with regards to the impact of data quality and CNR analysis:

1. The authors define the sequence-specific CNR as the difference between the mean values of GM and WM, divided by the sum of squares of their standard deviation. However, ideally, CNR values need to be corrected for differences in total acquisition time (T) as longer acquisition (such as for the MP2RAGE sequence) positively affect achievable CNR. As such, current CNR needs to be converted to CNR per unit time, by normalizing it to the square root of T (in minutes) as for example done in https://doi.org/10.3389/fnana.2016.00112. This, as well as the fact that the MP2RAGE T1w image is a combination of two separate inversion images, and thus based on more data, needs to be further discussed to put the results in perspective.

Response: Normalized CNR values - corrected per unit time similar to Haast et al. are given in the table below for both whole brain W/GM as well as DGM:

 CNRWdG CNRWG 

 Uncorrected Corrected* Uncorrected Corrected*

MPRAGE 0.16±1.00 0.05±0.34 2.52±0.33 0.80±0.10

MP2RAGE 0.98±0.13 0.30±0.04 3.53±0.12 0.97±0.04

 * Corrected = corrected per unit time similar to Haast et al.

As it can be observed from the Table, the general trend (higher CNR for the MP2RAGE compared to MPRAGE) was maintained. We have amended these normalized CNR value to the revised version of the manuscript in the corresponding sections of the Methods and results. 

2. Considering that the quality of the data might vary across the brain due to B1 transmit and receive homogeneity differences, as mentioned in the introduction by the authors too, I think it would be of interest to see whether difference in CNR between MPRAGE and MP2RAGE varies across the brain, and whether this spatial variation is more prominent for MPRAGE or MP2RAGE. This could be tested by averaging per lobe for example and will also ease interpretation of the observed differences between tissue probability maps.

Response: At 3T, increased rates of transmit and receive B1 in-homogeneities can create intensity variations in the acquired images leading to reduced contrast bias field. In MP2RAGE, by combining the images from the first and second readouts, T2* and B1 inhomogeneity effects can be largely canceled out, resulting in a strongly T1-weighted image with superior GM-to-WM contrast [Van de Moortele et al., 2009; Okubo et al., 2016]. Following the reviewers' request, we calculated the CNR differences between both acquired sequences within the different lobes. For MPRAGE; the calculated CNR values per lobe were [a.u]: 354.06±46.85, 398±48.46, 413.35±36.01, and 424.65±45.51 for the frontal, parietal, occipital, and temporal lobes respectively. In MP2RAGE, the calculated CNR per lobe were: 3053.64±52.24; 3132.97±49.32, 3133.63±49.96, and 3145.05±49.29 respectively. As it can be seen based on these values, in each modality, slight variations in CNR among the different lobes can be seen, however the range of variation of CNR relative to the group mean in MPRAGE (9-13%) is higher compared to MP2RAGE (1.5-1.7%). Thus, further strengthening the conclusion that despite local variation in CNR, T1-weighted image resulting from MP2RAGE, with superior GM-to-WM. In case the reviewer sees that this data would further strengthen our here reported results, we would add this gladly to the manuscript.

3. In line my previous comment and with the authors’ hypothesis stated in the

discussion that the observed differences in CNR “can be explained by the differences between the two sequences, and the fact that increased rates of bias-field effects are inherent in MP-RAGE images at high and ultra-high magnetic fields (≥3 Tesla), whereas, MP2RAGE is less sensitive to B1 bias, thus enabling the acquisition of images with enhanced” I would propose to calculate subject-specific difference maps between the MP-RAGE and MP2RAGE images (e.g., after normalization to the average CSF value, for example). This will provide a surrogate bias map and might provide additional information with regards to origin of the observed differences in terms of tissue segmentation. See also https://doi.org/10.3389/fnana.2016.00112 and https://doi.org/10.1016/j.neuroimage.2020.117373 for relevant demonstrations and discussions on the effect of B1 biases on cortical and subcortical segmentations using MP2RAGE data.

Response: We thank the reviewer for this suggestion. In the calculated intra-subject differences maps (see example below), we observe higher GM tissue probability in MP2RAGE compared to MP-RAGE, mainly in deep grey matter. This is consistent with the reported group-level VBM results in Figure 2. Moreover, for each subject, we calculated the dice similarity coefficient (DSC) in order to assess the similarity between GM and WM segmented maps achieved from both investigated sequences. The group mean DCS coefficient were as follows: DSCGM= 0.85±0.04; DSCWM= 0.94±0.01. These findings further confirm the values presented in Table 2 and Fig3, in which MP-RAGE resulted in higher whole-brain GM and DGM compared to MP-RAGE. Based on these findings, we strongly believe that the enhanced CNR of MP2RAGE enhances edge-detection during segmentation, leading to more accurate classification of the brain tissue types. 

Illustration of calculated difference tissue probability maps. Grey voxels represent voxels that were classified as GM by MP2RAGE only. Yellow voxels are voxels classified as GM in both sequences. 

With regards to the tissue probability map comparisons: 

4. Import details are lacking with regards to the voxel-based analyses of the tissue probability maps. The authors should elaborate more on the SPM analyses as it is impossible now for the reader to replicate the current analyses. For example, which registration tool was used? And why was chosen for a linear registration instead of non-linear transformation? The latter will ensure larger overlap across subjects for statistical analyses and more power. Moreover, did the authors co-register each MP-RAGE and MP2RAGE image individually to the template space or did they use the same transformation matrix for both types of images? This co-registration process should be clarified to rule out potential sources of biases in the comparison.

Response: For the VBM analyses, the obtained GM and WM probability maps from MP2RAGE and MP-RAGE using the different segmentation pipelines were registered to MNI using linear registration in SPM in order to achieve spatial overlap between subjects. Furthermore, VBM comparisons were performed using SPM software (SPM12; https://www.fil.ion.ucl.ac.uk/spm/software/spm12/). For the group-level voxel-wise comparisons, paired-samples t-tests were used for inter-modality comparisons (i.e, MP-RAGE vs. MP2RAGE) of whole-brain tissue probability maps obtained from the different segmentation tools. We stated this detail in the corresponding section of the Methods. 

Since the investigated participants in this study were all healthy without any history of any neurological disorders as well as no evidence of regional structural abnormalities and/or atrophy, we opted to use linear instead of non-linear registration. The use of non-linear registration might be advantageous in cases where there is large variance between the brain images from different subjects [Ying et al., 2017]. Moreover, since both images were acquired in the same MRI scan session, and using the same slice orientation and coverage for the MP-RAGE and MP2RAGE images were co-registered individually to MNI, in order to avoid un-necessary interpolations.

5. Finally, it is confusing why the WM results using volBrain are not shown in Figure 2 as for CAT-12 and FSL-FAST and as suggested in the caption.

Response: We agree with the reviewer. In order to maintain coherence in the displayed results in the figure, we modified Figure 2, adding the WM voxel-wise results. As mentioned in the results, no significant differences were found in WM using volBrain between both sequences. 

In general:

6. Especially in the introduction the authors should better emphasize the novelty of the current work (e.g., compared to https://doi.org/10.1002/jmri.24960).

Response: We thank the reviewer for this remark. In the referenced study by Okubo et al., 2016, the authors showed that compared to MP-RAGE, MP2RAGE showed higher reproducibility and CNR between tissue classes compared to MP-RAGE. In their study, the authors applied CAT12 (http://www.neuro.uni-jena.de/cat/) which relies on the segmentation algorithm embedded in SPM. In our study, we systematically investigated both the reproducibility and patterns of tissue classification differences between both sequences using other segmentation tools including CAT12, FSL-FAST, volBrain, and Freesurfer, which are widely used and available environments in use. Thus, enabling us to demonstrate whether a tool-sequences interaction can be detected. 

In order to better emphasize the novelty of this study, we modified the aims of the study paragraph as well as follows: '' in this study, we aimed to systematically examine the patterns of discrepancies and quantitatively assess the agreement between tissue classification obtained by different widely available automatic segmentation tools relying on different algorithms in the assessment of whole-brain tissue, as well as DGM volumes based on MP2RAGE compared to MP-RAGE images.'' 

7. I think the paper would benefit from an extra table/figure that summarizes all findings. This will make it easier for the reader to identify the key message of the manuscript and follow the narrative of the discussion.

Response: We agree with this suggestion of the esteemed reviewer. We have accordingly modified Figure 1 illustrating the steps conducted by the different used tools based on MP-RAGE and MP2RAGE sequences. Moreover, we added Figure 3; a scatter plots demonstrating the differences in tissue volumes measured by the different segmentation tools based on MP-RAGE and MP2RAGE sequences. 

Minor points

Introduction:

8. Define ‘high’ and ‘ultra-high’ magnetic field strengths

Response: it is commonly accepted that field strengths 1T-3T are defined as high-field, whereas field strengths of ≥7T are 'ultra-high'. We added this clarification to the corresponding part in the introduction as advised. 

9. Last paragraph, ‘doesn’t’ to ‘does not’

Response: This was corrected as suggested.

Test-retest reliability (methods):

10. Add characteristics for the test-retest subjects

Response: In the longitudinal group, N=12 participants (6 females, mean age ± SD = 51.5 ± 6.5 years) were included, and re-scanned after 1 year using the same MRI protocol. This information was now added to the corresponding section in the Methods.

11. First sentence: ‘described above’ to ‘described below’ or move paragraph

Response: This was corrected accordingly.

12. Second sentence:

a. Change ‘T1-MPRAGE’ to MP-RAGE to follow previous notations

b. Change ‘patients’ to ‘participants’

Response: Thank you for pointing out these points. These were changed accordingly.

MRI acquisition protocol (methods):

13. No need to define MP-RAGE again.

Response: This was adjusted as advised. 

14. Remove sequence parameters from text as this is redundant with Table 1. Also add readout bandwidth to better estimate SNR and interpret the CNR differences.

Response: We now added the readout bandwidth for both MP-RAGE (150 Hz/Px), and for MP2RAGE (240 Hz/Px) to Table 1.

15. Last sentence: would be informative for the reader to add a bit more detail what the authors are referring to with ‘Table 1 & Figure 1’

Response: Thank you for this suggestion. This sentence now reads as follows: '' See Table 1 for detailed sequences parameter and Figure 1 for examples illustrating MRI images acquired using both sequences'' allowing more details on what is depicted in both Table and Figure referred to. 

MRI data processing (methods):

16. As for MP-RAGE, no need to define CNR again.

Response: This was adjusted throughout accordingly.

Whole brain tissue segmentation (methods):

17. First sentence: change ‘free-widely’ to ‘free and widely’

18. Last sentence:

c. ‘Montreal neurological institute’ to ‘Montreal Neurological Institute’

Response: Both points pointed to above were adjusted accordingly in the corresponding section in the Methods.

Contrast-to-noise (methods):

19. Subscript ‘WG’ in ‘CNRWG’ as used in the equation and later in the results

Response: This was adjusted as suggested.

DGM segmentation (methods):

20. I do not understand why there is a difference in the tools used for whole-brain and DGM segmentation. I would suggest using the same set of tools across both analyses or motivate why it was chosen not to.

Response: In this study we aimed to systematically assess the differences in volumetric measures for whole brain and DGM between MP-RAGR and MP2RAGE using the different commonly-used automatic tools. Different tools and segmentation pipelines tools chosen by the users depending mainly on familiarity and prior experience. The main reason behind the discrepancy between the used tools across the different analyses, is to offer the readership a wide-and- comprehensive overview about the volumetric differences that can be observed between both modalities when assessed by the different tools, we opted to use open source tools operating on Unix-FSL, Matlab-SPM, as well as volBrain; is an open-access free of charge platform. As such, we included FSL FAST and FIRST, as well as volBrain for whole brain and DGM measures. In SPM, CAT-12 is mainly used for whole brain volumetric assessments and VBM analysis, however it is less used for DGM volumetric assessment. Moreover, it DGM volumetric assessment are often reported relying on Fressurfer as well. Therefore, in order to keep consistency in the number of used tools for whole-brain and DGM, we decided to include Freesurfer as third tool for DGM segmentation. In case the reviewer sees that this might make the paper more coherent and justified the Freesurfer DGM results can be omitted from the manuscript. We modified the aims paragraph in order to make this point more clearer as follows:'' we aimed to systematically examine the patterns of discrepancies and quantitatively assess the agreement between tissue classification obtained by different widely available, open-source automatic segmentation tools relying on different algorithms that are commonly used for the assessment of whole-brain tissue, as well as DGM volumes based on MP2RAGE compared to MP-RAGE images''. 

Statistical analysis (methods):

21. Is there are specific reason why the authors did not use a multivariate (i.e., across multiple regions of interest) ANOVA test for the statistical differences? This will also allow to test for the effect of segmentation tools, region of interest and/or potential interactions.

Response: We thank the reviewer for this remark. In this study, our main aims were to investigate the patterns of volumetric differences between MP-RAGE and MP2RAGE sequences using widely available methods, as well as examine the scan-rescan stability of both sequences. By adjusting for subject-related inherent characteristics, the used statistical test (Paired-samples t-test), both in the voxel-wise and volumetric comparisons, allows us to infer on the questions at hand.

We do strongly agree with the reviewer that an ANOVA model or multivariate analyses would allow to infer about possible effects of segmentation tool, or possible tool × sequence interactions. However, we refrained from using multivariate models in the current analysis due to two main reasons: Firstly, the relatively small sample size of the cross-sectional comparisons (N=29), and N=12 in the scan-rescan sub-study limits our statistical power, and the number of main effects and interactions we can explore. We highlighted this point as a limitation of the study. Secondly, since our overriding aim in this paper is to systematically demonstrate that due its' higher CNR and lower susceptibility to B1 inhomogeneity, MP2RAGE-derived measures are more sensitive and reproducible, inference regarding possible interactions were out of scope of the current study aims. We fear that drawing such conclusions would sound less impartial, and would possibly lead readers toward using a certain segmentation method, which we hope to avoid. Yet, this raised point remains very relevant to pursue in future follow-up studies, including larger numbers of participants.

Whole-brain volumes (results):

22. I think it would be helpful for the reader to see example tissue segmentation results for a single subject using both types of data. This could for example be combined with Figure 1.

Response: We have included a figure (Figure 1B) illustrating the tissue probability maps achieved by the different segmentation tools.

DGM segmentation (results):

23. Second sentence: ‘accumbnes’ to ‘accumbens’

Response: This was corrected.

Test-retest reliability (results):

24. First sentence: 

d. ‘MP2RGAE’ to ‘MP2RAGE’

e. Higher what? Volume? Please specify

Response: Thank you for pointing out these points. These were corrected. This sentence now reads as follows: '' At re-scan, MP2RAGE demonstrated higher whole-brain GM, lower WM, and CSF volumes compared to MP-RAGE in all three used tools.''

References

1. Coupé P, Manjón JV, Fonov V, Pruessner J, Robles M, Collins DL. Patch-based segmentation using expert priors: application to hippocampus and ventricle segmentation. Neuroimage. 2011;54(2):940-954. doi:10.1016/j.neuroimage.2010.09.018

2. Fischl B, Salat DH, Busa E, et al. Whole brain segmentation: automated labeling of neuroanatomical structures in the human brain. Neuron. 2002;33(3):341-355. https://surfer.nmr.mgh.harvard.edu/ftp/articles/fischl02-labeling.pdf

3. Haast RAM, Ivanov D, Formisano E, Uludaǧ K. Reproducibility and Reliability of Quantitative and Weighted T1 and T2∗ Mapping for Myelin-Based Cortical Parcellation at 7 Tesla. Front Neuroanat. 2016;10. doi:10.3389/fnana.2016.00112

4. Manjón JV, Tohka J, García-Martí G, et al. Robust MRI brain tissue parameter estimation by multistage outlier rejection. Magn Reson Med. 2008;59(4):866-873. doi:10.1002/mrm.21521

5. Manjón JV, Eskildsen SF, Coupé P, Romero JE, Collins DL, Robles M. Nonlocal intracranial cavity extraction. Int J Biomed Imaging. 2014;2014:820205. doi:10.1155/2014/820205

6. Manjón JV, Coupé P. volBrain: An Online MRI Brain Volumetry System. Front Neuroinform. 2016;10. doi:10.3389/fninf.2016.00030

7. O’Brien KR, Kober T, Hagmann P, et al. Robust T1-weighted structural brain imaging and morphometry at 7T using MP2RAGE. PLoS One. 2014;9(6):e99676. doi:10.1371/journal.pone.0099676

8. Okubo G, Okada T, Yamamoto A, et al. MP2RAGE for deep gray matter measurement of the brain: A comparative study with MPRAGE. J Magn Reson Imaging. 2016;43(1):55-62. doi:10.1002/jmri.24960

9. Patenaude B, Smith SM, Kennedy DN, Jenkinson M. A Bayesian model of shape and appearance for subcortical brain segmentation. Neuroimage. 2011;56(3):907-922. doi:10.1016/j.neuroimage.2011.02.046

10. Rajapakse JC, Giedd JN, Rapoport JL. Statistical approach to segmentation of single-channel cerebral MR images. IEEE Trans Med Imaging. 1997;16(2):176-186. doi:10.1109/42.563663

11. Smith SM. Fast robust automated brain extraction. Hum Brain Mapp. 2002;17(3):143-155. doi:10.1002/hbm.10062

12. Tustison NJ, Avants BB, Cook PA, et al. N4ITK: improved N3 bias correction. IEEE Trans Med Imaging. 2010;29(6):1310-1320. doi:10.1109/TMI.2010.2046908

13. Van de Moortele P-F, Auerbach EJ, Olman C, Yacoub E, Uğurbil K, Moeller S. T1 weighted brain images at 7 Tesla unbiased for Proton Density, T2* contrast and RF coil receive B1 sensitivity with simultaneous vessel visualization. Neuroimage. 2009;46(2):432-446. doi:10.1016/j.neuroimage.2009.02.009

14. Ying S, Li D, Xiao B, Peng Y, Du S, Xu M. Nonlinear image registration with bidirectional metric and reciprocal regularization. PLOS ONE. 2017;12(2):e0172432. doi:10.1371/journal.pone.0172432

---

## [Decision Letter · Decision Letter 1]

13 May 2021

Pécs, Hungary

May 13, 2021

PONE-D-20-40185R1

Whole brain and deep gray matter structure segmentation: quantitative comparison between MPRAGE and MP2RAGE sequences

PLOS ONE

Dear Dr. Droby,

Thank you for submitting your manuscript (R1 version) to PLOS ONE. After careful consideration, we feel that it has merit but does not fully meet PLOS ONE’s publication criteria as it currently stands. Therefore, we invite you to submit a revised version of the manuscript that addresses the points raised by Reviewer #2.

We look forward to receiving your revised manuscript.

Kind regards,

Joseph Najbauer, Ph.D.

Academic Editor

PLOS ONE

Journal Requirements:

Reviewers' comments:

Reviewer's Responses to Questions

**Comments to the Author**

1. If the authors have adequately addressed your comments raised in a previous round of review and you feel that this manuscript is now acceptable for publication, you may indicate that here to bypass the “Comments to the Author” section, enter your conflict of interest statement in the “Confidential to Editor” section, and submit your "Accept" recommendation.

Reviewer #1: All comments have been addressed

Reviewer #2: (No Response)

2. Is the manuscript technically sound, and do the data support the conclusions?

Reviewer #1: Yes

Reviewer #2: Yes

3. Has the statistical analysis been performed appropriately and rigorously? 

Reviewer #1: Yes

Reviewer #2: Yes

4. Have the authors made all data underlying the findings in their manuscript fully available?

Reviewer #1: Yes

Reviewer #2: No

5. Is the manuscript presented in an intelligible fashion and written in standard English?

Reviewer #1: Yes

Reviewer #2: No

6. Review Comments to the Author

Reviewer #1: I would like to thank the authors for addressing all my comments.

I have no additional comments regarding the revised manuscript.

Reviewer #2: I appreciate the authors' efforts to accommodate most of my concerns, which has improved the manuscript. I like the new Figure 3, which nicely shows the differences for the GM, WM and CSF segmentations between acquisitions for each method. WM and CSF are generally larger based on MPRAGE data using FSL-FAST (especially) and volBrain, while more similar using CAT12. The opposite is true for GM. In addition, I thank the authors for calculating CNR per lobe and I think this is informative and should be added to the main text.

Finally, I recommend further clarification of a couple of remaining issues:

- The authors state that 'MP2RAGE resulted in more accurate brain tissue segmentation' (abstract). This needs to be rephrased as the 'ground truth' is not known so it is difficult to define 'more accurate', or, it should be clarified what they mean with 'more accurate'.

- I think the difference between the MPRAGE and MP2RAGE sequences need to be emphasized more. Especially in terms of how the T1w-like image is obtained (i.e., synthesized) for the MP2RAGE sequence.

- In their response to my comment #3, the authors provided a intra-subject difference map based on the segmentation results. Although informative as well, what I was suggesting was to calculate the difference between the raw images. This allows characterization of any spatial bias in the difference between MPRAGE and MP2RAGE data. For example, are differences larger towards the frontal lobe?

- Please verify that CNR values were calculated using the raw T1w images, and elaborate on what the potential impact of readout bandwidth differences between MPRAGE and MP2RAGE acquisitions on CNR could be.

- I would advise to add the T-statistical value and degrees of freedom to the statistical results. Also, specify that the non-binary probability masks were used and if statistical testing was restricted to GM and/or WM (i.e., masked), or not.

Figures:

- Figure 1, if possible, I would recommend to use data from the same subject for all sub-panels.

- Figure 4, it is very difficult to see the significant voxels. I would advise to zoom in on the interesting part and only show that in the figure.

Some minor textual comments I noticed:

- Abstract: 'In a sub-study, twelve participants were scanned after one year.' change to ' rescanned'

- Abstract: 'Mean ± SD = 0.97 ± 0.04 and 0.8 ± 0.1 respectively; p<0.0001' change 'and' to 'vs'

- Introduction, last paragraph: 'different widely available, open-source automatic segmentation tools' remove open-source. I was not able to find the source code for volBrain

- Methods, CNRwg: 'corrected per scan time' change to 'normalized for scan time'

- Please go through the manuscript to correct small grammatical errors.

7. PLOS authors have the option to publish the peer review history of their article (what does this mean?). If published, this will include your full peer review and any attached files.

Reviewer #1: No

Reviewer #2: **Yes: **Roy Haast

---

## [Author Response · Author response to Decision Letter 1]

7 Jun 2021

Reply to Reviewer's comments

Reviewer #2: I appreciate the authors' efforts to accommodate most of my concerns, which has improved the manuscript. I like the new Figure 3, which nicely shows the differences for the GM, WM and CSF segmentations between acquisitions for each method. WM and CSF are generally larger based on MPRAGE data using FSL-FAST (especially) and volBrain, while more similar using CAT12. The opposite is true for GM. In addition, I thank the authors for calculating CNR per lobe and I think this is informative and should be added to the main text.

We thank the reviewer for his valuable efforts and constructive feedback and suggestions. We added the CNR per lobe to the text. Furthermore, we added the range of variation of CNR per lobe as suggested to the results section.

Finally, I recommend further clarification of a couple of remaining issues:

1. The authors state that 'MP2RAGE resulted in more accurate brain tissue segmentation' (abstract). This needs to be rephrased as the 'ground truth' is not known so it is difficult to define 'more accurate', or, it should be clarified what they mean with 'more accurate'.

Response: We thank the reviewers for this comment. Indeed, the use of the phrase "more accurate" here is not accurate. Therefore, we replaced it with "reproducible", as supported by the results of test- retest study. We also changed this in the conclusions paragraph in the discussion.

2. I think the difference between the MPRAGE and MP2RAGE sequences need to be emphasized more. Especially in terms of how the T1w-like image is obtained (i.e., synthesized) for the MP2RAGE sequence.

Response: We have modified the third paragraph describing MP2RAGE in the introduction, further clarifying how the MP2RAGE image is acquired and synthesized.

3. In their response to my comment #3, the authors provided an intra-subject difference map based on the segmentation results. Although informative as well, what I was suggesting was to calculate the difference between the raw images. This allows characterization of any spatial bias in the difference between MPRAGE and MP2RAGE data. For example, are differences larger towards the frontal lobe?

Response: We apologize for the misunderstanding of this point in the previous round. Since MPRAGE is not a quantitative sequence, and due to the difference in dynamic range between both sequences comparison of the raw images might not be meaningful (or accurate). Nevertheless, to perform the spatial comparison as suggested we first normalized both images relative to the white matter signal, and then we calculated the differences between the normalized images (See Figure Rev1. below). As it can be seen in the figure, following normalization only minor differences were demonstrated between the sequences for the WM and GM tissues (<±0.25, normalized values). 

Figure Rev1:

4. Please verify that CNR values were calculated using the raw T1w images, and elaborate on what the potential impact of readout bandwidth differences between MPRAGE and MP2RAGE acquisitions on CNR could be.

Response: We confirm that CNR was calculated on the acquired raw T1 images. We clarified this point in the corresponding section in the methods. MP2RAGE has higher bandwidth compared to MPRAGE (240 Hz/Px vs. 150 Hz/Px). The use of high bandwidth is expected to reduce susceptibility effects including eddy currents associated with metal, and thus to improve image SNR [See: Marquess et al., 2010, Neuroimage]. We added this information to the discussion.

5. I would advise to add the T-statistical value and degrees of freedom to the statistical results. Also, specify that the non-binary probability masks were used and if statistical testing was restricted to GM and/or WM (i.e., masked), or not.

Response: We now added the T statistical values in the corresponding sections in the results as suggested. Also, in Statistical analysis sub-section of the Methods, we section further clarified that the statistical testing was performed on the masked GM and WM probability maps achieved by the different used segmentation tools.

Figures:

6. Figure 1, if possible, I would recommend to use data from the same subject for all sub-panels.

Response: All the images in the sub-panels in Figure are actually from the same subject. As noted by the reviewer, slight variations in the slices were found however, we now rectified this and made sure that all slices match. 

7. Figure 4, it is very difficult to see the significant voxels. I would advise to zoom in on the interesting part and only show that in the figure.

Response: We agree with this comment. We have reduced the number of slices shown in the left panel to better depict the significant voxels in GM and WM.

Some minor textual comments I noticed:

8. Abstract: 'In a sub-study, twelve participants were scanned after one year.' change to ' rescanned'

Response: This was changed as suggested.

9. Abstract: 'Mean ± SD = 0.97 ± 0.04 and 0.8 ± 0.1 respectively; p<0.0001' change 'and' to 'vs'

Response: The word and was changes to VS.

10. Introduction, last paragraph: 'different widely available, open-source automatic segmentation tools' remove open-source. I was not able to find the source code for volBrain

Response: We omitted the phrase "open-source" as advised.

11. Methods, CNRwg: 'corrected per scan time' change to 'normalized for scan time'

Response: This was changed accordingly.

12. Please go through the manuscript to correct small grammatical errors.

Response: We went over the manuscript thoroughly once again and corrected some minor mistakes. Hopefully we got them all!

---

## [Decision Letter · Decision Letter 2]

30 Jun 2021

Pécs, Hungary

June 29, 2021

Whole brain and deep gray matter structure segmentation: quantitative comparison between MPRAGE and MP2RAGE sequences

PONE-D-20-40185R2

Dear Dr. Droby,

We’re pleased to inform you that your manuscript (R2 version) has been judged scientifically suitable for publication and will be formally accepted for publication once it meets all outstanding technical requirements.

PLEASE NOTE:

References 1, and 10 are not complete (the last names of the authors are missing, also the year of publication and the journal volume and page naumbers are missing - PLEASE CORRECT.

PLEASE CHECK ALL REFERENECES FOR COMPLETENESS AND ACCURACY.

PLEASE CHECK THE ENTIRE MANUSCRIPT FOR COMPLETENESS AND ACCURACY.

Kind regards,

Joseph Najbauer, Ph.D.

Academic Editor

PLOS ONE

Reviewers' comments:

Reviewer's Responses to Questions

**Comments to the Author**

1. If the authors have adequately addressed your comments raised in a previous round of review and you feel that this manuscript is now acceptable for publication, you may indicate that here to bypass the “Comments to the Author” section, enter your conflict of interest statement in the “Confidential to Editor” section, and submit your "Accept" recommendation.

Reviewer #2: All comments have been addressed

2. Is the manuscript technically sound, and do the data support the conclusions?

Reviewer #2: Yes

3. Has the statistical analysis been performed appropriately and rigorously? 

Reviewer #2: Yes

4. Have the authors made all data underlying the findings in their manuscript fully available?

Reviewer #2: No

5. Is the manuscript presented in an intelligible fashion and written in standard English?

Reviewer #2: Yes

6. Review Comments to the Author

Reviewer #2: (No Response)

7. PLOS authors have the option to publish the peer review history of their article (what does this mean?). If published, this will include your full peer review and any attached files.

Reviewer #2: **Yes: **Roy AM Haast

---

## [Editor Report · Acceptance letter]

21 Jul 2021

PONE-D-20-40185R2 

Whole brain and deep gray matter structure segmentation: quantitative comparison between MPRAGE and MP2RAGE sequences 

Dear Dr. Droby:

I'm pleased to inform you that your manuscript has been deemed suitable for publication in PLOS ONE. Congratulations! Your manuscript is now with our production department. 

Kind regards, 

on behalf of

Dr. Joseph Najbauer 

Academic Editor

PLOS ONE